# Parameter Optimization and Potential Bioactivity Evaluation of a Betulin Extract from White Birch Bark

**DOI:** 10.3390/plants9030392

**Published:** 2020-03-23

**Authors:** Haiyan Chen, Han Xiao, Jiwei Pang

**Affiliations:** 1Changchun Sci-Tech University, Changchun 130600, Jilin, China; pjw980222@163.com; 2College of Food Science and Engineering, Jilin Agriculture University, Changchun 130118, Jilin, China; ybestxh@163.com

**Keywords:** betulin, ultrasonic-assisted ethanol extraction, optimization, bioactivity, *Betula platyphylla* Suk. bark

## Abstract

Owing to its pharmacological potential, betulin has attracted substantial attention in the past two decades. The present work attempts to extract betulin from *Betula platyphylla* Suk. bark by the ultrasonic-assisted ethanol method and to evaluate its potential bioactivities. The critical process variables affecting the yield were optimized by a four-factor, three-level, central composite response surface methodology (RSM). A betulin yield of 92.67% was achieved under the optimum conditions: 65% ethanol concentration, 1:25 ratio of white birch bark to solvent, an extraction temperature of 30 °C, and an extraction time of 30 min. The ratio of solid to solvent is the most significant parameter in terms of yield. The optimal conditions were validated through experiments, and the observed value (92.67 ± 2.3%) was interrelated with the predicted value (92.86 ± 1.5%). The betulin extract was analyzed quantitatively by HPLC and quantitatively by LC/MS, before its potential biological activities were evaluated. Bioactivity surveys confirmed that the betulin extract showed not only no embryo deformity through zebrafish administration experiments, but also no cytotoxicity through MTT assays. Furthermore, the betulin extract had strong antioxidant activities in vitro by scavenging ferric reducing power (FRAP), 1,1-diphenyl-2-picryl hydrazyl(DPPH), 2,2′-azinobis-(3-ethylbenzothiazoline-6-sulfonic acid) (ABTS), and chelating metal ions. This study demonstrates that ultrasonic-assisted ethanol extraction may be a green, efficient method for the extraction of betulin from white birch bark, and that betulin extracts are potentially useful in cosmetics, food supplements, or pharmaceutical applications.

## 1. Introduction

As a large-scale timber-processing byproduct, birch tree bark is a natural source of lupane-type triterpenoid betulin, whose concentration accounts for approximately 30 wt% (mass weight percentage) [1]. Many researchers have demonstrated that betulin and its derivates show useful properties for treating metabolic disorders, infectious diseases, cardiovascular disorders, and neurological disorders [2]. For instance, Narala and co-workers investigated the significant scavenging activity of betulin and cystone against DPPH, NO, and superoxide radicals in comparison to standard antioxidant L-ascorbate (L-AA) [3]. Srisombat and co-workers isolated betulinc acid from *Cratoxylum formosum subsp. pruniflorum* (*Kurz*) and evaluated its antioxidant and anti-HIV activities [4]. Yang and co-workers investigated the antioxidant capacity of betulin in PC12 cells against H_2_O_2_-induced cytotoxicity in vitro; the IC_50_ value of betulin was shown to be 6.88 μg/mL [5]. As a consequence, betulin extraction from birch tree bark has attracted research interest all over the world [6]. Therefore, it is highly desirable to develop a facile and green extraction approach for the betulin active compound in birch tree bark. This has been viewed as a promising way to valorize birch tree bark byproducts.

To date, numerous methods for the extraction of the betulin active compound from birch tree bark have been reported in the literature. State-of-the-art extraction technologies result in excellent betulin yield, i.e., ranging from 20% to 40% from the birch outer bark [7]. For instance, Rizhikovs et al. reported that approximately 37% of triterpene-rich extracts were isolated from outer birch bark by hot water and Na_2_CO_3_ pretreatment [8]. Chen et al. investigated ultrasonic-assisted 98% ethanol solvent extraction for betulin from white birch bark, and optimized the extraction parameters using a response surface methodology. Under the optimal conditions, 23.17% of the betulin product was obtained [9]. Sergey et al. developed a four-step processing technique to obtain a betulin yield of 51% from birch (*Betula pendula*) bark [10]. Recently, subcritical water extraction for betulinic acid from birch bark has been proposed, and 2.803 mg/g birch bark of betulinic acid was achieved under the optimal conditions [11,12]. 

Among reported extraction methods, the most widely used technique involves ethanol solvent extraction. On the one hand, ethanol is relatively less toxic than acetone and methanol. On the other hand, the ethanol extraction method is more cost-effective in comparison with subcritical/supercritical extraction, because the latter approaches are always undertaken under high pressure [11,12]. However, there is limited information on the toxicity and antioxidant activities of betulin extracts from the bark of *Betula platyphylla* Suk. in the literature. In the present work, ultrasonic-assisted ethanol extraction is investigated, and the critical parameters affecting the betulin yield are optimized using a response surface methodology (RSM). The betulin extract was preliminarily purified through AB-8 macroporous absorption resin, and the betulin concentration was then determined by high performance liquid chromatography (HPLC) and its relative molecular weight mass was measured by a liquid chromatography/mass spectrometer (LC-MS). Finally, the potential bioactivities of the betulin extract were tentatively evaluated through the embryo deformity of zebrafish, HaCaT cell proliferation, and in vitro anti-oxidation activities of scavenging FRAP, DPPH, ABTS, and chelating metal ions. 

## 2. Materials and Methods

### 2.1. Materials

White birch (*Betula platyphylla* Suk.) bark was purchased from a local market in Changbaishan, Jilin Province, in the northeast of China. The bark samples were immediately dried at 40 °C for 12–18 h and stored in a dry and dark place. All organic solvents, i.e., methanol, ethanol, t-butanol, amyl alcohol, glacial acetic acid, and perchloric acid, used in this work were of analytical grade (purity ≧ 98%) and were bought from Tianjin Fuyu Fine Chemical Co., Ltd. (Tianjin, China). The betulin standard (purity ≧ 99%) was obtained from Chengdu Manst Biotechnology Co., Ltd. (Chengdu, China). Vanillin (purity ≧ 99%) was bought from Tianjin Guangfu Technology development Co., Ltd. (Tianjin, China). AB-8 mesoporous absorption resin was obtained from Shandong Donghong Chemical Co., Ltd. (Shandong, China). The DPPH radical was purchased from Sigma Aldrich (St. Louis, Missouri, MO, USA).

### 2.2. Betulin Extraction from Birch Bark by Ultrasonic-Assisted Ethanol Solvent

Single-factor experiments and a central composite design were conducted to optimize the critical parameters for betulin extraction from birch bark at a fixed ultrasonic frequency of 50 kHz and an ultrasonic input power of 400 W (SK-14GT ultrasonic equipment, Shanghai, China). The extraction temperature was digitally controlled in a HH-2 thermo water bath (Zhengzhou, Henan province, China). The total content of betulin in the extract was determined by a UV-Vis spectrometer in operational parameter optimization experiments because of its simple and fast characteristics. The betulin concentration after purification was quantitively detected by HPLC in our work.

For single-factor experiments, five critical factors affecting the betulin yield were investigated, i.e., the solvent type (water, methanol, ethanol, butanol, and amyl alcohol), ethanol concentration (45%, 55%, 65%, 75%, and 85%), extraction time (20, 25, 30, 35, and 40 min), extraction temperature (20, 30, 40, 50, and 60 °C), and solid/solvent ratio (1:20, 1:25, 1:30, 1:35, and 1:40 mg/mL). In the single-factor experiment design, only one factor was varied at a time, while the other factors were kept constant in each experiment [13]. For each experimental test, 5 g of dried outer bark of *Betula platyphylla* Suk. birch was grounded and put in a three-neck flask. Once extraction was complete, the sample was filtered under a vacuum to achieve a crude betulin extract. The belutin yield was calculated according to Equation (1).
(1)$$Betulin\;yield\;(\%)=\frac{Weight\;of\;betulin\;extract\;from\;bark\;(g)}{Weight\;of\;bark\;material\;(g)}\times 100\%$$

For the central composite design (CCD) experiment, the dependent response was the betulin yield, and the five independent variables were the solvent type, ethanol concentration, extraction time, temperature, and solid/solvent ratio. A four-factor, three-level test with 29 trials was conducted to investigate the interaction effect between two factors on the belutin yield [14]. Table 1 shows the range and level of the four variables. Based on the CCD experimental data, an empirical second-order polynomial equation model involving the betulin yield (Y) was speculated, according to Equation (2):(2)$$Y=\beta_{0}\;+\;\sum \beta_{i}\;X_{i}\;+\;\sum \beta_{\mathrm{ii}}X_{i}^{2}\;+\;\sum \sum \beta_{\mathrm{ij}}X_{i}X_{j},$$
where *Y* represents the betulin yield (dependent response), *β*_0_ is the constant coefficient, *β*_i_*, β*_ii_, and *β*_ij_ are the coefficients for the linear, quadratic, and interaction terms, respectively, and *X*_i_ and *X*_j_ represent the corresponding independent variables. Subscript i and j are the numbers 1 to 4. 

The Design-Expert software (Version6.0.10, Stat-Ease. Inc., Minneapolis, MN, USA) was used to fit the CCD experimental data. From the determination coefficient (*R*^2^) and the analysis of variance (ANOVA), the reliability of the regression model could be evaluated. Furthermore, when one independent variance was kept constant while other variables were changed, the contour plots and three-dimensional (3D) response surfaces were attained to yield the optimum parameters.

### 2.3. Quantitative and Quantitative Analyses of the Betulin Extract

#### 2.3.1. Purification of the Betulin Extract

The crude betulin extract was preliminarily purified through AB-8 macroporous absorption resin using EtOAc as the eluent phase [15]. The eluent was collected and evaporated at 40 °C under a vacuum of −0.1 MPa. After being dried at 40 °C for 12 h in a vacuum drier, the purified betulin sample was obtained for the following analyses.

#### 2.3.2. UV-Vis Analysis

During parameter optimization, the total concentration of betulin in the extract was quantitatively measured by Uvmini-1280 UV-vis spectroscopy (Shimadzu (Suzhou) Co., Ltd., China) [16]. Before sample determination, the betulin standard curve at 551 nm was established as y = 6.5989 x + 0.0156 (*R*^2^ = 0.9992), where y represents the betulin concentration (mg/mL) and x represents the OD_210_ value. In the range of 0~0.3 mg/mL of betulin concentration, a good linear calibration fit was achieved. *R*^2^ = 0.9992 indicates good linearity. According to the betulin standard cure, the betulin concentration in the extract was calculated. All measurements were performed in triplicate, and the final results were expressed as the average value ± SD. 

#### 2.3.3. HPLC Analysis

After purification, the betulin concentration in the extract was determined by HPLC on a Series 1500 HPLC system (Alltech, KY, USA) with a C18 column (4.6 mm × 250 mm, 5 µm) [17]. The mixture solution of methanol/water (85:15, *v/v*) was used as mobile phase, with a flow rate of 1.0 mL/min. The mobile phase was filtered through a 0.45 µm membrane filter, and then degassed ultrasonically before use. Other conditions were a sample injection volume of 20 µL, a column temperature of 35 °C, and a UV detection wavelength of 204 nm.

#### 2.3.4. LC-MS Analysis

The betulin extract was qualitatively analyzed by liquid chromatography/mass spectroscopy (LC/MS) [18]. The betulin solution (100 μg/mL in methanol) was filtered through a 0.45 μm hydrophobic PTFE filter and analyzed by an LC/MS Waters analytical system (Aquity^TM^ Waters, USA) equipped with a photodiode array (PDA) detector and Quattro Premier XE electrospray ionization (ESI) mass spectrometer. A Hypersil Gold aQ column (50 mm × 2.1 mm, 3.0 μm) was used and the column temperature was set to 40 °C. The sample injection volume was 10 μL. The mobile phase consisting of aqueous formic acid [0.5% (*v/v*)] (A) and methanol (B) was delivered at the flow rate of 0.25 mL/min with the programmed gradient elution presented in Table 2.

### 2.4. Toxicity and Antioxidant Bioactivities of the Betulin Extract

#### 2.4.1. Acute Toxicity Assessment of the Betulin Extract

The acute toxicity of the betulin extract was investigated according to the procedures reported by Chen et al. [19]. Briefly, the acute toxicity of the betulin extract was assessed by the abnormal phenotypes and mortality of Zebrafish through oral experiments. Zebrafish wild type (WT) embryos (*n* = 30) 6 hours post fertilization (6 hpf) and larvae (*n* = 30) 3 days post fertilization (3 dpf) were used as the administered groups in this work. Different betulin concentrations were set to 100, 1000, and 2500 μg/mL, labeled as low, medium, and high concentration groups, respectively. The abnormal phenotypes and mortality of each treated group were documented for 72 h sequential observation. The blank controls were Zebrafish WT embryo and larvae (*n* = 30) incubated in the breeding water. Each experiment was repeated three times.

#### 2.4.2. Cytotoxity Assay of the Betulin Extract

Cytotoxity measurement of the betulin extract was performed using a 3-(4, 5-dimethylthiazol-2-yl)-2,5-diphenyl tetrazolium bromide (MTT) assay, according to the procedures detailed by Kpemissi and co-workers [20]. Briefly, HaCaT cells were first seeded at 1.0 × 10^4^ cells/well in 96-well microtiter plates in DMEM medium (with 10% FBS and 1% P/S) and incubated in 5% CO_2_ atmosphere at 37 °C for 24 h. Then, 200 μL betulin extract (dissolved in the DMEM medium with a concentration of 16.625–125 μg/mL) was transferred to the wells, followed by 24 h incubation, and the MTT solution with a final concentration of 0.5 mg/mL was then added to 96-well microtiter plates. After incubation for 4 h, the medium was aspirated by filter paper, the purple crystals were dissolved in DMSO, and the absorbance of the resulting solution in each well was measured at 570 nm with a microtiter plate reader (Varioskan Flash, Thermo Scientific, Waltham, MA, USA). The viability of cells in the medium only including betulin standard with a concentration of 100 μg/mL was used as the positive control. The viability of cells without a betulin extract drug was used as the negative control. Each experiment was repeated three times. 

#### 2.4.3. In Vitro Antioxidant Activities of the Betulin Extract

##### Metal Chelating Activity

The chelating assay of the betulin extract was performed according to the method reported by Katak et al. [21]. Briefly, 50 μL of betulin extract (125–1000 μg/mL) was incubated with 5 μL of ferrous chloride (2 mM). After 5 min, the reaction was initiated by adding 15 μL of ferrozine (5 mM). The mixture solution was incubated at room temperature for 10 min. The absorbance of the ferrous ion-ferrozine complex was determined by UV-Vis spectra at 562 nm. The ability of the betulin extract to chelate ferrous ion was calculated according to Equation (3). The chelating assay of the betulin standard with a concentration of 150 μg/mL was also measured as the control.
(3)$$Chelating\;activity\;(\%)=\frac{OD_{control}-OD_{sample}}{OD_{control}}\times 100,$$
where OD_control_ and OD_sample_ represent the absorbance (OD_562_) of the ferrous ion-ferrozine complex without and with the betulin extract, respectively. 

##### 2,2′-Azinobis-(3-ethylbenzothiazoline-6-sulfonic acid) (ABTS) Radical Scavenging Activity

The ABTS scavenging activity of the betulin extract was determined according to the procedure described by Zhang et al. [18]. Briefly, stock solutions consisting of ABTS solution and potassium persulfate solutions were prepared. An amount of stock solution was mixed with betulin extract and then incubated for 2 h. Afterwards, the absorbance of the mixture solution was determined by UV-Vis spectra at 735 nm. The ABTS radical scavenging activity of the betulin extract was calculated according to Equation (4). The ABTS scavenging activity of the betulin standard with a concentration of 150 μg/mL was also measured as the control.
(4)$$ABTS\;radical\;scavenging\;activity\;(\%)=\frac{OD_{control}-OD_{sample}}{OD_{control}}\times 100,$$
where OD_control_ and OD_sample_ represent the absorbance (OD_735_) of stock solution without and with the betulin extract, respectively. 

##### Ferric Reducing Power (FRAP) Activity

The FRAP activity of the betulin extract was investigated according to the method detailed by Tálos-Nebehaj and co-workers [22]. Briefly, a fresh working solution was prepared by mixing acetate buffer, 2,4,6-tri(2-pyridyl)-s- triazine (TPTZ), and FeCl_3_·6H_2_O, and then kept at 37 °C until use. The betulin extract was reacted with FRAP solution in the dark at room temperature for 30 min. Afterwards, the reaction solution was withdrawn and detected with UV-Vis at 595 nm. The FRAP activity was calculated according to Equation (5). The FRAP activity of the betulin standard with a concentration of 150 μg/mL was also measured as the control.
(5)$$ABTS\;radical\;scavenging\;activity\;(\%)=\frac{OD_{control}-OD_{sample}}{OD_{control}}\times 100,$$
where OD_control_ and OD_sample_ are the absorbance (OD_595_) of the mixture solution without and with the betulin extract, respectively. 

##### Determination of the DPPH (1,1-Diphenyl-2-picryl hydrazyl) Radical Scavenging Activity

The DHHP radical scavenging activity of the betulin extract was examined according to the method detailed by Zhang and co-workers [23]. Briefly, different concentrations of betulin extract (125–1000 μg/mL) in methanol were mixed with an equal volume of 100 μM DPPH in methanol and the mixture was kept in the dark for 30 min. The absorbance was detected at 517 nm with a UV-Vis spectrophotometer. The percent DPPH scavenging activity was calculated according to Equation (6). The DHHP radical scavenging activity of the betulin standard with a concentration of 150 μg/mL was also measured as the control.
(6)$$DHHP\;radical\;scavenging\;activity\;(\%)=\frac{OD_{control}-OD_{sample}}{OD_{control}}\times 100,$$
where OD_control_ and OD_sample_ are the absorbance (OD_517_) of the mixture solution without and with the betulin extract, respectively. 

### 2.5. Statistical Analysis

The statistical analyses were performed using SPSS 22.0 (IBM, Armonk, NY, USA), and the final data were presented as means ± SD. All experiments were conducted three times and the value *p* < 0.05 was accepted as the level of statistical significance.

## 3. Results and Discussion

### 3.1. Single-Factor Experiment Design for Betulin Extraction by the Ultrasonic-Assisted Ethanol Method

The solvent most suitable for betulin extraction was chosen by screening five types of solvents, i.e., methanol, ethanol, *n*-butanol, pentanol, and H_2_O. The concentration of organosolvent was set at 65%. The ratio of solid to solvent was set at 1:25. The experimental results in Figure 1A show that the maximal betulin yield of 90.25% was obtained by ethanol, followed by methanol (81%), pentanol (72%), and butanol (64.25%). When H_2_O was used as the solvent, only a 55.25% betulin yield was obtained. A significant difference (*p* < 0.05) in the betulin yield was observed between ethanol extraction and extraction using the other solvents. Therefore, in terms of the betulin yield, ethanol was chosen as the extraction solvent in the following experiment.

Then, single-factor experiments were designed to examine the influence of four critical parameters, i.e., the ethanol concentration, the ratio of bark to solvent, the extraction temperature, and the extraction time, on the betulin yield from white birch bark using the ethanol extraction method. In order to reduce the extraction time and increase the extraction rate, ultrasonic-assisted technology was employed during the whole process [21]. The results are depicted in Figure 1B–D.

As seen in Figure 1B, approximately an 85% betulin yield was obtained after 30 min extraction. When expanding the extraction time beyond 30 min, the betulin yield remained almost stable up to 35 min and then decreased greatly (*p* < 0.05) to a 60%. This was probably due to the fact that other compounds were extracted along with betulin when the extraction time was elongated to 40 min [24]. As detailed in Figure 1C, the betulin extraction yield depending on the solid/solvent ratio was investigated. It could be observed that the betulin yield increased with the enhancement of the solid/solvent ratio, reaching a maximal betulin yield (87.75%) at a solid/solvent ratio of 1:30. However, no significant difference (*p* < 0.05) in the betulin yield was observed for the solid/solvent ratio factor. The impact of the extraction temperature on the betulin yield is depicted in Figure 1D, which demonstrates that the highest yield of betulin (90%) was obtained at 45 °C. An extraction temperature at 45 °C was accompanied by a decrease of the betulin yield; at 65 °C, the yield decreased further (*p* < 0.05) to approximately 62%. The effect of the ethanol concentration on the betulin yield was investigated; the results are shown in Figure 1E. The maximal betulin yield (87.25%) was achieved using 65% ethanol extraction, and the betulin yield remained almost stable with increasing the ethanol concentration up to 85%. In other words, 65% ethanol is sufficient to obtain a high betulin yield. Based on the data of single-factor experiments, the variables of an extraction time of 25–35 min, extraction temperature of 20–40 °C, ethanol concentration of 55–75%, and solid/solvent ratio of 20:1–30:1 were adopted for the following RSM parameter optimization.

### 3.2. RSM Optimization for Betulin Extraction

Central composite RSM was performed using four factors and three levels with 29 trials to evaluate the interaction effect of the four independent variables (i.e., extraction time, ethanol concentration, extraction temperature, and solid/solvent ratio) on the betulin yield. It can be observed from Table 3 that the betulin yield ranged from 86.50% to 92.75% in this work. Multiple regression analysis was used to fit the RSM experimental data, and a quadratic polynomial model was achieved by employing Equation (7), which describes the relationship between the betulin yield (Y) and the independent variables.
Y (%) = 92.75 + 0.17X_1_ + 0.65X_2_ + 0.50X_3_ + 0.10X_4_ + 1.75 X_1_ X_2_ − 0.25X_1_X_3_ − 0.37X_1_X_4_ − 0.44 X_2_ X_3_ + 0.25 X_2_ X_4_ − 1.19 X_3_ X_4_ − 1.79 X_1_^2^ − 2.14 X_2_^2^ − 1.35 X_3_^2^ − 1.14 X_4_^2^,(7)
where Y is the betulin yield (%), and X_1_, X_2_, X_3_, and X_4_ are the ethanol concentration, solid/solvent ratio, temperature, and time, respectively. 

In the quadratic polynomial model, negative coefficients indicate an unfavorable effect on the betulin yield, while positive coefficients show a favorable effect [25]. As demonstrated by the values of the coefficient in Equation (7), the influence of the four independent variables on the betulin yield decreased by the order of solid/solvent ratio (0.65) > temperature (0.50) > ethanol concentration (0.17) > extraction time (0.10).

The significance and reliability of the quadratic polynomial model (Equation (7)) were evaluated by an analysis of variance (ANOVA); the values are shown in Table 4.

The *F*- and *p*-values are always expected to be important indicators for ANOVA; the higher the *F*-value and the smaller the *p*-value, the more significant the quadratic polynomial model [23]. In Table 4, the *F*-value of the model is 20.30, indicating that the probability of such a model is more than 99.99% from the point of view of the *p*-value (0.0001); that is, the quadratic polynomial model is highly significant and reliable, and can be used to predict the betulin yield. In addition, the mean square (0.37) and *p*-value (<0.0001) of the lack of fit are both very low, demonstrating the goodness-of-fit and the suitability of the quadratic polynomial model. Furthermore, the goodness-of-fit of the model was also confirmed by the calculated multiple correlation co-efficient (*R*^2^ = 0.9536) and adjusted determination co-efficient (*R*^2^_adj_ = 0.9061). This suggests that 95.36% of the variations were demonstrated by the model, and/or 90.61% of the total variations were illustrated by the model. The *R*^2^_adj_ value is smaller than the *R*^2^ value due to there being more than three terms in the model [26]. As indicated by the *p*-values in Table 4, the *p*-values of the linear co-efficient (X_2_, X_3_), interaction term co-efficient (X_1_X_2_, X_3_X_4_), and quadratic term co-efficient (X_1__→4_^2^) are highly significant (*p* < 0.01), indicating that these significant factors have a strong influence on the betulin yield.

Figure 2 shows the interaction effects between two independent variables on betulin yield. The apex of the three-dimensional (3D) picture presents the maximal value of the predicted yield, and the corresponding point is visualized by the smallest ellipse in the bottom contour diagram of the 2D contour plot [25,26]. For instance, the interactive effects between the ethanol concentration and the solid/solvent ratio are shown in Figure 2A, which is a typical saddle-shaped picture. Clearly, the apex of the picture is the maximal value of the betulin yield of ~92%. However, a further increase of one or both of the variables results in an adverse effect on the betulin yield, for which the lowest value is approximately 88%. Similar phenomena were observed in Figure 2B–F. The optimal conditions of betulin extraction obtained by the ethanol method are shown in Figure 2. The optimal parameters in real operational experiments were as follows: 65% ethanol concentration, 1:25 solid/solvent ratio, 32 °C temperature, and 30 min extraction time. Under the aforementioned optimized conditions, a 92.86 ± 1.5% predicted betulin yield (*Y*) was obtained from the regression model.

The optimal condition was validated through three repeated experiments under the optimized conditions. The results show that the observed value of the betulin yield is 92.67 ± 2.3%, which is only 0.09% different from the predicted betulin yield (92.86 ± 1.5%). This indicates that the regression model can be used to predict the betulin extract yield in the future.

### 3.3. Qualitative and Quantitative Analyses of the Betulin Extract 

Before HPLC analysis, the crude betulin extract was preliminarily purified through AB-8 macroporous absorption resin. The betulin concentration after purification was quantitatively determined by HPLC; the results are shown in Figure 3A,B. It can be observed that apart from the main betulin, with a yield of 92.67%, a total of 7.33% of other pentacyclic triterpenes was also extracted, i.e., betulinic acid, oleanolic acid, and lupeol (Figure 3B). Tálos-Nebehaj and co-workers reported that the tree outer bark consists of more bioactive compounds than the inner bark, and that these bioactive compounds showed high antioxidant activities [18]. It was reported that these birch triterpenes showed antiviral, antimicrobial, and hepatoprotective pharmacological activities [12,13,15]. According to the standard curve of betulin (Figure 3C), the betulin concentration in the extract sample was calculated to be 50%. Šiman and co-workers developed ethanol extraction following five purification steps to obtain high purity betulin (≥99%) from birch bark [15]. In addition, by using LC/MS, it was revealed that the molecular mass weight of betulin was 442.7 g/mol (M+H^+^); the chemical molecular structure is presented in Figure 3D. To date, the methods of HPLC and LC-MS have been widely used for qualitative and quantitative analyses of betulin and other triterpenes in extract samples [17,27,28]. In our work, the purification of betulin in the extract has not yet been sufficiently completed, and study is ongoing in our lab. 

### 3.4. Acute Toxicity of the Betulin Extract 

The acute toxicity of the betulin extract was investigated through Zebrafish administration experiments in this work because zebrafish oral experiments used to evaluate the toxicity of bioactive compounds are widely available in the literature [19,29]. In the orally administered groups, Zebrafish embryos were treated by betulin extracts with concentrations of 100, 1000, and 2500 μg/mL, respectively. The experiments were divided into two groups: control groups (or normal groups) and treatment groups. After treatment for a assigned period (8–72 h), the zebrafish were then removed from the dish, and their shape and body color were observed under a microscope to evaluate the abnormal phenotypes in terms of the following three aspects [19]: First, the body becomes transparent and yellow; second, the body length shortens; and third, the axis bends and the embryonic notochord is seriously twisted into an “S” shape. According to the abnormal phenotypes, we were able to evaluate the acute toxicity of the betulin extract. As shown in Figure 4, the shape and body color of the zebrafish showed no obvious change before and after treatment. This implies that the betulin extract has a negligible acute toxicity in zebrafish embryonic development. The conclusion was therefore drawn that betulin extracted from birch bark by ultrasonic-assisted ethanol solution is safe. The betulin extract has promising potential in food, cosmetic, and pharmaceutical applications. 

### 3.5. HaCaT Cell Viability of the Betulin Extract

The cytotoxic effect of the betulin extract when using different alcohol concentrations (55–75% ethanol and 65% pentanol) on HaCaT cell viability was investigated using the MTT assay; the results are shown in Figure 5. It can be observed that betulin extract and betulin standard show no cytotoxic effects against HaCaT cell growth, and their significance effect (*p* < 0.05) is found in comparison with the blank control. This indicates that the betulin extract has a small cytotoxicity effect on HaCaT cell viability. The concentration-response bar charts in Figure 5 show that HaCaT cells significantly grow well at 15.625 μg/mL of betulin extract. Furthermore, the betulin extract treated by 65% ethanol exerts a higher cell viability than 65% pentanol because of the different compounds in the extract by two kinds of solvents. Simultaneously, the effect of the betulin extract when using different alcohol concentrations on HaCaT cell viability is not significant (*p* < 0.05) in comparison with the betulin control. This phenomenon was in good agreement with reports in the literature [2,3,4,5,30], in which it was demonstrated that the betulin extract from birch treated by ether exerted a lower inhibition of cell viability than that treated by supercritical fluid ethanol (SFE) or exudates. Szuster-Ciesielska and Kandefer-Szerszeñ reported that betulin was a bioactive protectant of HepG2 cells against ethanol-induced cytotoxicity [31]. Lee and co-workers revealed that betulin could protect Hom cells cultured in a medium. A betulin extract with a concentration of 15.625 μg/mL could protect T-22 hippocampal cell viability against reactive oxygen species-stressed cytotoxicity [32]. As a result of the zebrafish experiments, it was concluded that betulin extract has negative cytotoxicity and shows potential for use in food and pharmaceutical applications.

### 3.6. In Vitro Antioxidant Activities of the Betulin Extract

In vitro antioxidant activities of the betulin extract were determined using FRAP, chelating metal ion, DPPH, and ABTS assays; the results are shown in Figure 6. Generally, the antioxidant activities of the betulin extract treated by 55% ethanol were significantly higher than those with 65% and 75% ethanol (*p* < 0.05). Simultaneously, the in vitro antioxidant activities of the belutin extract showed a concentration-dependent tendency. However, 150 μg/mL of the betulin standard shows higher antioxidant capacities than the betulin extract. This can probably be attributed to the unfavorable effect of impurity compounds in the extract. Hofmann et al. reported that each antioxidant activity assay is specific to certain types of antioxidants, and none of the assays evaluate the overall antioxidant capacity of a plant extract [32]. Therefore, combined multi-assay methods for antioxidant activity evaluation may be propitious to pursuing complex antioxidant properties in betulin extracts. Nalara and co-workers demonstrated the significant scavenging activity of betulin against DPPH, nitric oxide, and superoxide radicals [3]. Zhang and co-workers reported that when the concentration of betulin was more than 2000 μg/mL, the scavenging rate of ABTS and DPPH radicals reached over 90% [23]. In addition, Blondeau and co-workers reported that extracts from white birch showed strong antimicrobial properties with a minimal inhibitory concentration, i.e., between 0.83 and 1.67 mg/mL, and catechol was viewed as one of the main components contributing to the antimicrobial activity of this extract [33]. These findings reveal that betulin has significant antioxidant abilities in vitro and is viewed as a good preservative in the food industry. 

## 4. Conclusions

In summary, we investigated and optimized the parameters of ultrasonic-assisted ethanol extraction for betulin from birch bark through single-factor experiments and RSM design. A betulin yield of 92.67% was obtained under the optimized extraction conditions. This was validated through three repeated experiments and the observed value (92.67 ± 2.3%) was well interrelated with the predicted value (92.86 ± 1.5%). The betulin concentration in the extract was determined to be 50% by HPLC. Apart from the main betulin with the yield of 92.67%, other pentacyclic triterpenes comprising 7.33% of the extract were detected. Furthermore, the potential bioactivities of the betulin extract, i.e., toxicity and in vitro antioxidant activities, were evaluated. The results showed that the betulin extract exhibited no embryo deformity and no cytotoxicity, as well as high in vitro antioxidant activities. It can be concluded that betulin extract obtained by the ultrasonic-assisted ethanol approach is a promising potential candidate for food and pharmaceutical applications. 

## Figures and Tables

**Figure 1 plants-09-00392-f001:**
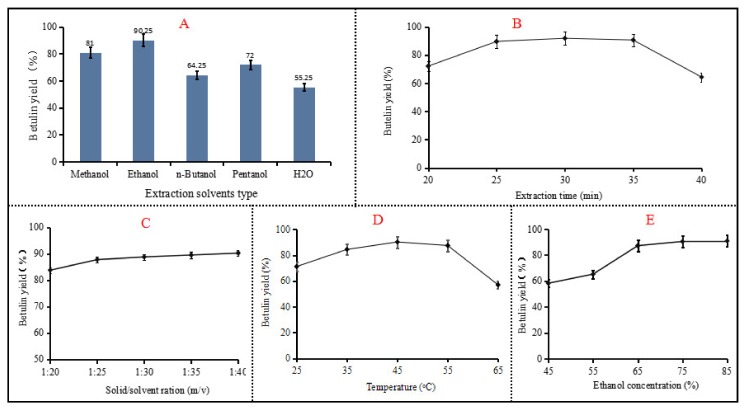
Influences of different extraction process variables on the betulin yield (**A**: extraction solvent type; **B**: extraction time; **C**: solid/solvent ratio; **D**: extraction temperature; **E**: ethanol concentration).

**Figure 2 plants-09-00392-f002:**
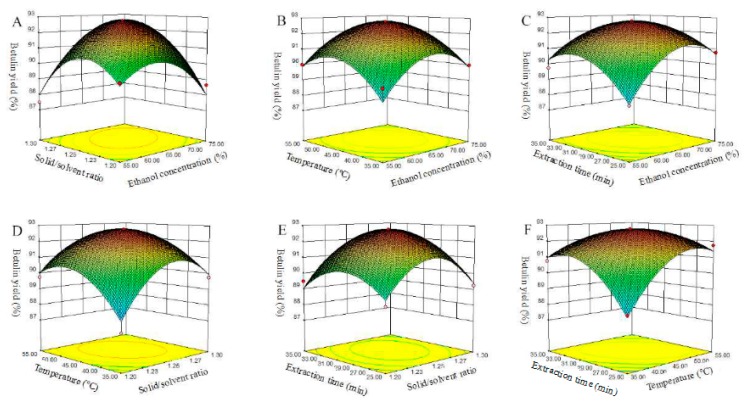
Response surface plots related to the interactive effects of the ethanol concentration (X1), solid/solvent ratio (X2), temperature (X3), and experiment time (X4) on the betulin yield. (**A**) interaction of solid/solvent ratio and ethanol concentration; (**B**) interaction of temperature and ethanol concentraction; (**C**) interaction of extraction time and ethanol concentraction; (**D**) interaction of temperature and solid/solvent ratio; (**E**) interaction of extraction time and solid/solvent ratio; (**F**) interaction of extraction time and temperature.

**Figure 3 plants-09-00392-f003:**
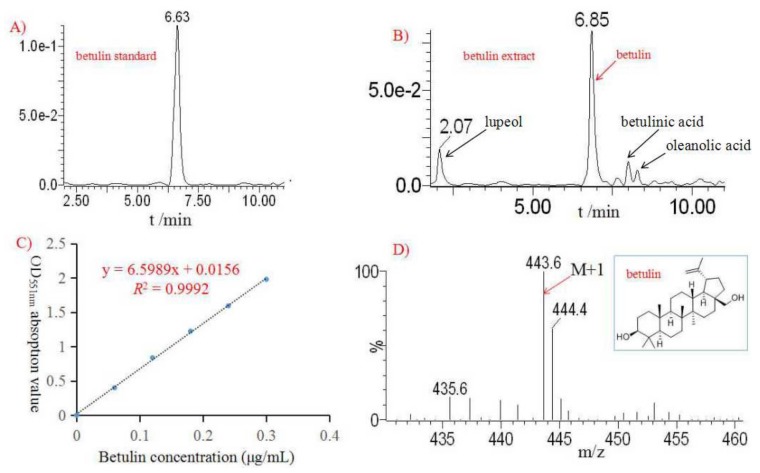
Qualitative and quantitative analyses of betulin extract conducted by high performance liquid chromatography (HPLC) (**A**,**B** and **C**,**D**) and LC/MS (**D**).

**Figure 4 plants-09-00392-f004:**
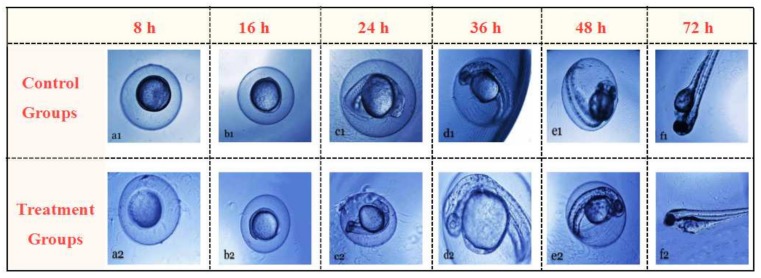
Effect of the betulin extract on zebrafish embryotoxicity (**a1**–**f1**) control groups (or normal groups) and (**a2**–**f2**) treatment groups. (**a1**,**a2**) 8 h—90%-epiboly; (**b1**,**b2**) 12 h—the first body node has a front boundary at this time, and the eye primordium begins to appear; (**c1**,**c2**) 24 h—primordium-phase 5, where the telencephalon is significant on the dorsal side of the front end of the nerve axis; (**d1**,**d2**) 36 h—primordium-25, where the pigment extends near the end of the tail; (**e1**,**e2**) 48 h—long-pectoral fin period, where the finned buds bend backward and have obvious sharp angles; (**f1**,**f2**) 72 h—burst period, where embryo hatching is basically complete and the anterior edge of the sternal fin continues to expand.

**Figure 5 plants-09-00392-f005:**
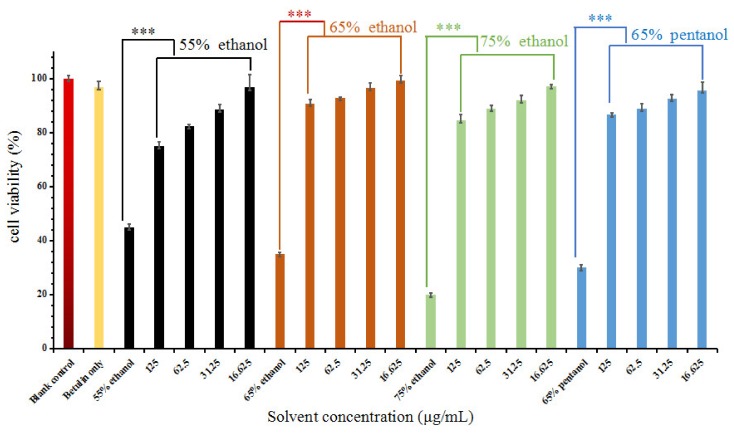
Cell viability assay of the betulin extract with different ethanol concentrations (*** significance *p* < 0.05, The concentration of betulin standard was set at 100 μg/mL).

**Figure 6 plants-09-00392-f006:**
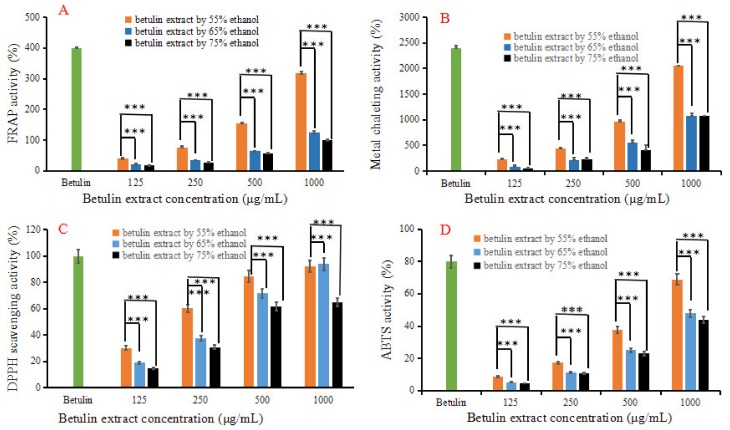
In vitro antioxidant activities of the betulin extract. (**A**) PRAP activity; (**B**) Metal chelating assay, (**C**) DPPH scavenging activity; (**D**) ABTS radical scavenging activity assay. *** means significantly different (*p* < 0.05) in the same concentration. The concentration of betulin standard was set at 150 μg/mL.

**Table 1 plants-09-00392-t001:** The design of central composite design (CCD) experiments.

Independent Variables		Levels		
		−1	0	+1
1	Ethanol concentration (%)	55	65	75
2	Solid/solvent ratio (mg/mL)	1:20	1:25	1:30
3	Extraction temperature (°C)	20	30	40
4	Extraction time (min)	25	30	35

**Table 2 plants-09-00392-t002:** Programmed gradient elution of mobile phase for liquid chromatography/mass spectroscopy (LC/MS) analysis.

Time (min)	Ratio of A (%)	Ratio of B (%)
0–3.0	30	70
3.0–4.0	30–10	70–90
4.0–5.0	10–2	90–98
5.0–12.0	2	98
12.0–12.1	2–30	98–70
12.1–15.0	30	70

**Table 3 plants-09-00392-t003:** Central composite response surface methodology (RSM) and experimental data.

Trials	X_1_: Ethanol Concentration/%	X_2_: Solid/Solvent Ratio (g/mL)	X_3_: Temperature/°C	X_4_:Extraction Time/min	Y: Betulin Yield/%
1	0	1	0	−1	89.25
2	0	−1	−1	0	87.25
3	1	1	0	0	91.25
4	0	−1	0	1	89.50
5	0	0	0	0	92.75
6	1	0	−1	0	90.00
7	0	0	0	0	92.75
8	0	1	−1	0	89.75
9	0	−1	0	−1	88.75
10	0	1	1	0	90.50
11	−1	0	0	1	89.75
12	−1	0	−1	0	89.25
13	0	0	0	0	92.75
14	1	0	0	−1	90.75
15	−1	1	0	0	87.25
16	0	0	0	0	92.75
17	-1	0	1	0	90.00
18	0	0	1	−1	91.75
19	0	0	−1	−1	88.25
20	0	0	0	0	92.75
21	−1	−1	0	0	89.50
22	1	0	0	1	89.25
23	1	−1	0	0	86.50
24	0	0	−1	1	90.75
25	0	−1	1	0	89.75
26	0	1	0	1	91.00
27	0	0	1	1	89.50
28	−1	0	0	−1	89.75
29	1	0	1	0	89.75

**Table 4 plants-09-00392-t004:** Analysis of variance (ANOVA) of the quadratic polynomial model.

Source	Sum of Squares	Degrees of Freedom	Mean Square	*F* Value	*p*-Value
Model	75.58	14	5.40	20.30	<0.0001
X_1_	0.33	1	0.33	1.25	0.2818
X_2_	5.01	1	5.01	18.82	0.0007
X_3_	3.00	1	3.00	11.28	0.0047
X_4_	0.13	1	0.13	0.49	0.4956
X_1_X_2_	12.25	1	12.25	46.05	<0.0001
X_1_X_3_	0.25	1	0.25	0.94	0.3488
X_1_X_4_	0.56	1	0.56	2.11	0.1679
X_2_X_3_	0.77	1	0.77	2.88	0.1119
X_2_X_4_	0.25	1	0.25	0.94	0.3488
X_3_X_4_	5.64	1	5.64	21.21	0.0004
X_1_^2^	20.82	1	20.82	78.28	<0.0001
X_2_^2^	29.58	1	29.58	111.20	<0.0001
X_3_^2^	11.89	1	11.89	44.72	<0.0001
X_4_^2^	8.36	1	8.36	31.44	<0.0001
Residual	3.72	14	0.27		
Lack of fit	3.72	10	0.37		<0.0001
Pure error	0.000	4	0.000		
Total	79.30	28			
Multiple correlation co-efficient (*R*^2^)			0.9536		
Adjusted determination co-efficient (*R*^2^_adj_)			0.9061		

Note: significant, *p* < 0.05; highly significant, *p* < 0.01.

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
