# Peer review of "Parameter Optimization and Potential Bioactivity Evaluation of a Betulin Extract from White Birch Bark"

_plants, 2020, doi:10.3390/plants9030392_

Round 1

Reviewer 1 Report

The authors mainly focused on asserting the non-toxicity and safety of the betulin extract, although the biological efficacy in preclinical models should be first assessed. It would be interesting to investigate the mechanism by which betulin extract affects cell viability, resulting the data on biological efficacy very limited and incomplete. Therefore, the novelty of the work is limited and no significant addition to the existing knowledge on the topic has been made.

Despite the authors have made the requested changes in the text, they did not perform the required additional experiments and did not correct the figure. Overall, the data are not sufficient and the study needs further experimental tests.

For these reasons, my decision is to reject the manuscript.

-the authors have identified the minor components of the betulin extract, as required. However, to attribute the biological activity only to betulin, the other pentacyclic triterpenes should be tested on HaCaT cell viability, as already required. Indeed, the biological activity could not be due only to betulin, but a possible additive or synergistic effect of a pool of components of the extract could be verified. Despite the literature about the biological activity of pentacyclic triterpenes, a further experimental assay is needed to test the pentacyclic triterpenes present in your betulin extract to confirm the literature data and exclude the contribute to the biological activity of the extract.

-the figure 5 is incomplete. The effect of solvent without the betulin extract on HaCaT cell viability should be shown in figure 5, as already required. Also, the statistical significance should be shown in figure 5.

- the title of the work concerns the bioactivity of the betulin extract, but the reported data are not sufficient to demonstrate it. The in vitro antioxidant assays are not sufficient for the functional and biological characterization of the extract. The ability of the betulin extract to interfere with the intracellular Reactive Oxygen Species (ROS) should be tested in HaCaT cell lines, to validate the reported in vitro antioxidant activity. The measurement of intracellular ROS following treatment with the betulin extract should be performed, evaluating if the reduction of cell viability could be due to a potential modulation of the cell oxidative state by the betulin extract.

Author Response

-the authors have identified the minor components of the betulin extract, as required. However, to attribute the biological activity only to betulin, the other pentacyclic triterpenes should be tested on HaCaT cell viability, as already required. Indeed, the biological activity could not be due only to betulin, but a possible additive or synergistic effect of a pool of components of the extract could be verified. Despite the literature about the biological activity of pentacyclic triterpenes, a further experimental assay is needed to test the pentacyclic triterpenes present in your betulin extract to confirm the literature data and exclude the contribute to the biological activity of the extract.

Response: Thank you for your kind comments. This present work is focusing on the biological activities of the betulin extract, not the isolated ndividual component in the betulin extract. To my opinion, reviewer 1 expand the scope of our manuscript, the biological activity of individual component in the betulin extract is not belonged  to our manuscript, it is another research project in future.

-the figure 5 is incomplete. The effect of solvent without the betulin extract on HaCaT cell viability should be shown in figure 5, as already required. Also, the statistical significance should be shown in figure 5.

Response: The Figure 5 is revised. The effect of solvent without the betulin extract on HaCaT cell viability is added in figure 5. Also, the statistical significance is shown in figure 5.

-the title of the work concerns the bioactivity of the betulin extract, but the reported data are not sufficient to demonstrate it. The in vitro antioxidant assays are not sufficient for the functional and biological characterization of the extract. The ability of the betulin extract to interfere with the intracellular Reactive Oxygen Species (ROS) should be tested in HaCaT cell lines, to validate the reported in vitro antioxidant activity. The measurement of intracellular ROS following treatment with the betulin extract should be performed, evaluating if the reduction of cell viability could be due to a potential modulation of the cell oxidative state by the betulin extract.

Response: As similar as the comment 1, the ROS modulation of the HaCaT cell oxidative state by the betulin extract is also beyond from our manuscript content. It is belonging to another research project.

Reviewer 2 Report

The authors responded to all reviewer comments. The manuscript has been improved and can be recommended for publication.

Author Response

-The authors responded to all reviewer comments. The manuscript has been improved and can be recommended for publication.

Response: Thank you for your good comments. The authors have carefully re-checked the English spelling through out the whole manuscript.

Reviewer 3 Report

The manuscript entitled “Parameters Optimization and Bioactivities Evaluation of Betulin Extract from White Birch Bark” by Haiyan and Han intends to optimize the extraction of botulin using different conditions by RSM. The manuscript is interesting and in the Scopus of the journal. However, the English should be completely revised. Major concerns:

  • Why some words are in red color?
  • Line 20: “The betulin extract is quantitative and quantitative analyses by HPLC and LC/MS” please rephrase
  • Line 21 24: rephrase
  • Line 34: wt?
  • Line 54: “On the other hand, ethanol extraction method is more practical than subcritical water extraction due to operational cost and safety” I do not agree. Why? Please use references to reply.
  • Line 56: “antiooxidant" correct
  • Line 68: Why the authors dried at 55°C instead of 40ºC (to preserve antioxidants)?
  • In my opinion, the total concentration of botulin should be quantified by HPLC
  • Line 207/208: rephrase
  • The yield graphic should have statistical analyses
  • Line 219/220: why the yield may decrease if the extraction time increase? Explain.
  • 288: “quantitative and quantitative analyses” ?
  • 293/294: re phrase
  • 333: see mistakes
  • Cell studies: all text should be revised. Indeed, the graph should include statistical analysis.
  • Antioxidant activity should have statistical analysis.
  • In any part of the manuscript the authors mention if they used a UAE probe or bath. Also the equipment details should be presented. This is extremely important.
  • Conclusion should be revised.

Author Response

  1. Why some words are in red color?

Response: Thank you for your good suggestions. The words marked in red color has been corrected into black color.

  1. Line 20: “The betulin extract is quantitative and quantitative analyses by HPLC and LC/MS” please rephrase.

Response: The sentences has been corrected to “The betulin extract is analyzed quantitatively by HPLC and quantitatively by LC/MS...”

  1. Line 21-24: rephrase

Response: The sentences has been corrected to “The biological activity surveys have confirmed that the betulin extract not only shows no embryo deformity through zebrafish administration experiments, but also displays no cytotoxicity through MTT methods. Futhermore, the betulin extract has strong antioxidant activities in vitro by scavenging FRAP, DPPH, ABTS and chelating metal ions.”

  1. Line 34: wt?

Response: wt? is corrected to wt.% (mass weight percentage)

  1. Line 54: “On the other hand, ethanol extraction method is more practical than subcritical water extraction due to operational cost and safety” I do not agree. Why? Please use references to reply.

Response: The sentence has been corrected to “On the other hand, ethanol extraction method is more cost-effective than subcritical/supercritical extraction methods because the latter approaches are always operated under high pressures [see references 8 and 9].”

  1. Line 56: “antiooxidant" correct

Response: “antiooxidant" is corrected to “antioxidant”.

  1. Line 68: Why the authors dried at 55°C instead of 40ºC (to preserve antioxidants)? In my opinion, the total concentration of botulin should be quantified by HPLC.

Response: After the authors checked the experimental notes, they found that there is a spelling mistake. The sample was dried at 40°C for 12-18 h to avoid antioxidants loss at high temperature.

In addition, the total concentration of betulin is quantified by UV-Vis spectroscopy in the operational parameters optimization experiments because of its simple and fast characteristics, but the purity of betulin in the extract is quantitively detected by HPLC in our work.

  1. Line 207/208: rephrase

Response: The sentence is corrected to “H2O using as solvent, only 55.25% of betulin yield was obtained.” in lines 198-199 in the revised version.

  1. The yield graphic should have statistical analyses

Response: The yield graphic has been statistical analyses by SPSS software. And significant difference (p≦0.05) of betulin yield is observed between ethanol extraction and other solvents extraction. Other factors (including extraction time, solid/solvent ratio, temperature, and ethanol concentration) affecting betulin yield are also statistically analyzed. All revisiona are added in the revised version.

  1. Line 219/220: why the yield may decrease if the extraction time increase? Explain.

Response: As seen in Fig. 1B, approx. 85% of betulin yield is obtained after 30 min extraction. When expanding the extraction time beyond 30 min, the betulin yield keeps almost stable up to 35 min and then decrease greatly (p≦0.05) to 60% of betulin yield. It is probably contributed to the fact that other compounds are extracted along with betulin once the extraction time is enlonged to 40 min [Moorthy G., Maran J.P., Ilakya S., Anitha S.L., PoojaSabarima S., Priya B. Ultrasound assisted extraction of pectin from waste Artocarpus heterophyllus fruit peel [J]. Ultrason. Sonochem., 2017, 34: 525-530].

  1. Line 288: “quantitative and quantitative analyses” ?

Response: “quantitative and quantitative analyses” is corrected to “Prior to HPLC analysis” in line 279 in the revised version.

  1. Line 293/294: rephrase

Response: The sentence is corrected to “the mass weight of betulin in the extract was detected by LC/MS method, it reveals that betulin’s mass weight is 442.7 g/mol (M+H+)” in lines 288-290 in the revised version.

  1. Line 333: see mistakes

Cell studies: all text should be revised. Indeed, the graph should include statistical analysis.

Response: The whole text on cell studies has been revised. And the graph of Fig. 5 is included statistical analysis in the revised version.

  1. Antioxidant activity should have statistical analysis.

Response: Antioxidant activity in the graph of Fig. 6 is included statistical analysis in the revised version.

  1. In any part of the manuscript the authors mention if they used a UAE probe or bath. Also the equipment details should be presented. This is extremely important.

Response: The extraction temperature was digitally controlled in a HH-2 thermo water bath (Zhengzhou, Henan province, China) in lines 78-79.

  1. Conclusion should be revised.

Response: The conclusion has been revised.

Reviewer 4 Report

Comments:

    In this manuscript, the authors described “Parameters Optimization and Bioactivities Evaluation of Betulin Extract from White Birch Bark. This review reveals that this study has demonstrated that ultrasonic-assisted ethanol extraction may be a great efficiency proposal for betulin active compounds from white birch bark. The work seems well-done in the area. However, several points need to be clarified.

Comments for the Attention of the Author(s)

There are a few points that should be checked as follows:

  1. In the introduction, the author should add a description of betulin and its pharmacological activity.
  2. Cell viability assay of betulin extract with different ethanol concentration. Authors should add betulin only as a control group.
  3. In vitro antioxidant activities of betulin extract at different concentrations of ethanol. Authors should add betulin only as a control group.

Author Response

  1. In the introduction, the author should add a description of betulin and its pharmacological activity.

Response: Thank you for your good comments. In the introduction, a description of betulin and its pharmacological activity has been added in lines 36-42 in the revised version.

  1. Cell viability assay of betulin extract with different ethanol concentration. Authors should add betulin only as a control group.

Response: Cell viability assay of betulin standard was added as a control group in Fig. 5 in the revised version.

  1. In vitro antioxidant activities of betulin extract at different concentrations of ethanol. Authors should add betulin only as a control group.

Response: In vitro antioxidant activities of betulin standard with the concentration of 150 μg/mL was added as a control group in Fig. 6 in the revised version.

Reviewer 5 Report

This manuscript describes in a good way and in the form of a comprehensive study, the extraction of betulin from Betula platyphylla bark by ultrasonic-assited ethanol method and the evaluation of its in vitro bioactivities. 

The topic is very interesting for the use of natural bioactive molecules as cosmetics, food supplements or pharmaceutical research and the approach that even includes many in vitro assays is convincing.
The reviewer thinks that this work will be of interest for the readership of Plants and significantly contributes to the development of the field. Several papers on betulin bioactivity have been published, but this work is one of the very few approaches in which the bioactivity of the molecule is well characterized and correlated to the extraction method used.

Author Response

-This manuscript describes in a good way and in the form of a comprehensive study, the extraction of betulin from Betula platyphylla bark by ultrasonic-assited ethanol method and the evaluation of its in vitro bioactivities. 

The topic is very interesting for the use of natural bioactive molecules as cosmetics, food supplements or pharmaceutical research and the approach that even includes many in vitro assays is convincing.
The reviewer thinks that this work will be of interest for the readership of Plants and significantly contributes to the development of the field. Several papers on betulin bioactivity have been published, but this work is one of the very few approaches in which the bioactivity of the molecule is well characterized and correlated to the extraction method used.

Response: Thank you for your good comments. The authors have carefully re-chekced the whole manuscript to correct the existed bad sentences and grammatical errors.

Round 2

Reviewer 1 Report

The evaluation of the biological activity of the extract in the present work is still insufficient. The authors report that "The biological activity surveys have confirmed that the betulin extract not only shows no embryo deformity through zebrafish administration experiments but also displays no cytotoxicity" asserting that the extract is safe, i.e. does not induce cytotoxicity, on two different experimental models. However, the fact that the extract is safe doesn't mean that the extract has a biological activity, for example in terms of antioxidant activity on cells, anti-proliferative or protective against damage to cells, etc.

In this paper, the experimental approach to the evaluation of biological activity is not adequate, since the evaluation of the safety of an extract represents a consequential step to the validation of its biological efficacy. Here, no data on the biological activity of the extract on the cells is reported. The authors say that the extract is safe, but it does not mean that the extract is effective and active and the authors report in the line 344 that "extract can be served as a bioactive natural candidate for treating neurodegenerative diseases", resulting absolutely outside the possibility of this work based on the showed data.

The experimental approach to design Figure 5 is unclear. The authors do not justify the choice to evaluate the toxicity of ethanol on HaCaT cells, being keratinocyte cell line from adult human skin.

Furthermore, the authors show in Figure 5 the toxicity of ethanol respect to betulin extract but the point of the untreated cells is not shown in Figure 5. Indeed, in line 334 the authors say that “the effect of the betulin extract on HaCaT cell viability is not significant (p<0.05) in comparison with only betulin control” but the effect of betulin extract alone to untreated cells is not shown.

Furthermore, if the authors' goal is to evaluate the protective effect of the extract against the toxicity of ethanol, the authors do not explain why those concentrations of solvent were chosen, nor a dose-effect experiment for the solvent toxicity on this cell line was performed. The statistical analysis performed in Figure 5 is not clearly explained, neither in Materials and Methods nor in the caption of Figure 5. It is not clear the significance of respect to which point is considered.

Author Response

1. The evaluation of the biological activity of the extract in the present work is still insufficient. The authors report that "The biological activity surveys have confirmed that the betulin extract not only shows no embryo deformity through zebrafish administration experiments but also displays no cytotoxicity" asserting that the extract is safe, i.e. does not induce cytotoxicity, on two different experimental models. However, the fact that the extract is safe doesn't mean that the extract has a biological activity, for example in terms of antioxidant activity on cells, anti-proliferative or protective against damage to cells, etc.

Response: Thank you for your good comments. The authors are agreeing with you that betulin extract is safe based on the experimental data. But it does not mean that the betulin extract has a full biological activities, including antioxidant activity, anti-proliferative and cytotoxicity activities. In our present work, we have preliminarily investigated the antioxidant activities in vitro of betulin extract  by scavenging FRAP, DPPH, ABTS and chelating metal ions. However, the anti-proliferative and cytotoxicity activities of betulin extract are beyond from our manuscript. Maybe we will study them in future. 

2. In this paper, the experimental approach to the evaluation of biological activity is not adequate, since the evaluation of the safety of an extract represents a consequential step to the validation of its biological efficacy. Here, no data on the biological activity of the extract on the cells is reported. The authors say that the extract is safe, but it does not mean that the extract is effective and active and the authors report in the line 344 that "extract can be served as a bioactive natural candidate for treating neurodegenerative diseases", resulting absolutely outside the possibility of this work based on the showed data.

Response: As mentioned before, the biological activities of anti-proliferative and cytotoxicity activities against betulin extract are absolutely outside the scope of this work. The sentence in the line 344 has been corrected to " In combination of zebrafish experiments, a reasonable conclusion has been drawn that betulin extract has potential functions in food and pharmaceutical application due to its safety. " in the revised version.

3. The experimental approach to design Figure 5 is unclear. The authors do not justify the choice to evaluate the toxicity of ethanol on HaCaT cells, being keratinocyte cell line from adult human skin.

Response: The experimental approach about Figure 5 is only to show the cytotoxicity against the betulin extract no matter what cells is chosen.

4. Furthermore, the authors show in Figure 5 the toxicity of ethanol respect to betulin extract but the point of the untreated cells is not shown in Figure 5. Indeed, in line 334 the authors say that “the effect of the betulin extract on HaCaT cell viability is not significant (p<0.05) in comparison with only betulin control” but the effect of betulin extract alone to untreated cells is not shown.

Response:Figure 5 has been corrected and the column bars of the untreated cells (bland control) and betulin standard are shown in Fig. 5 in the revised version.

5. Furthermore, if the authors' goal is to evaluate the protective effect of the extract against the toxicity of ethanol, the authors do not explain why those concentrations of solvent were chosen, nor a dose-effect experiment for the solvent toxicity on this cell line was performed. The statistical analysis performed in Figure 5 is not clearly explained, neither in Materials and Methods nor in the caption of Figure 5. It is not clear the significance of respect to which point is considered.

Response: Different compounds have different solubility in different concentration of alcohols solvents. We would like to compare the potential functions of those betulin extract by 55% ethanol, 65% ethanol, 75% ethanol and 65% pentanol. However, the dose-effect experiments for the toxicity against cell viability are not conducted because those extracts are  mixture compounds. The statistical analysis performed in Figure 5 has been corrected in the revised version.

Reviewer 3 Report

The manuscript has been improved by the authors regarding some of my concerns. However, in my opinion, the cell assays discussion needs to be revised in what concerns to the English as well as the conclusion section.

Author Response

1. The manuscript has been improved by the authors regarding some of my concerns. However, in my opinion, the cell assays discussion needs to be revised in what concerns to the English as well as the conclusion section.

Response: Thank you for your good comments. The cell assays discussion have been corrected in the revised version. And the authors have carefully checked the English writing throughout the manuscript. All revisions are marked in Red.

Reviewer 4 Report

In the Figure 5, the author cannot set the betulin control to 100%. Betulin is a positive control. The blank group without drugs is set to 100%.

Author Response

1. In the Figure 5, the author cannot set the betulin control to 100%. Betulin is a positive control. The blank group without drugs is set to 100%.

Response: Thank you for your good comments. Figure 5 has been corrected. The blank control without drugs is set to 100% and its column bar has been added in Figure 5.

Round 3

Reviewer 3 Report

The authors improved the english but, in my opinion, the cell assays section could still be improved. It is up to the Editor decision.

Author Response

1. The authors improved the English but, in my opinion, the cell assays section could still be improved. It is up to the Editor decision.

Response: Thank you for your good comments. We are taking a good account into your comments, and carefully checking the issues involving English writing and cell assays section. Some  bad sentences in this section are deleted and we rephrase this section in lines 33-349 in the revised version. We are hoping that this round revision is suitable for acceptance.

This manuscript is a resubmission of an earlier submission. The following is a list of the peer review reports and author responses from that submission.

Round 1

Reviewer 1 Report

The article is original.

The presented research data are important scientifically and practically.

I recommend publishing. 

Reviewer 2 Report

The manuscript entitled: „Parameters Optimization and Bioactivities Evaluation of Betulin Extract from White Birch Bark„ show some new findings, worth publication. The paper is written in good style with provided an explanation and reason for conducted research. The methods are described in detail. Other parts of the manuscript are also well described and discussed. The only what I missing it has been concluded by the authors in the conclusion part: “In future, we will identify the chemical compositions of betulin extract, and elucidate the main components contributing to the cytotoxicity and antioxidant bioactivities of this betulin extract.” In my opinion, that information about the composition of the obtained extract, not only about the presence of one compound “botulin” should be added to the present manuscript. The addition of such information would improve the scientific quality of the manuscript. It needs to be also highlighted, that in the present form the manuscript it seems that it is methodological paper.

Reviewer 3 Report

Manuscript Number: Plants 732724

Title: Parameters Optimization and Bioactivities Evaluation of Betulin Extract from White Birch Bark

In the work the authors aimed to optimize the extraction parameters of betulin extract from white birch bark (Betula platyphlly) and evaluate the biological activity.

Several scientific papers about the different extraction methods of betulin from white birch bark are already published. Chen QH et al., 2009 previously reported the “Optimization of ultrasonic-assisted extraction (UAE) of betulin from white birch bark”. Moreover, the authors mainly focused on asserting the non-toxicity and safety of the betulin extract, although the biological efficacy in preclinical models should be first assessed. It would be interesting to investigate the mechanism by which betulin extract affects cell viability, resulting the data on biological efficacy very limited and incomplete.

Therefore, the novelty of the work is limited and no significant addition to the existing knowledge on the topic has been made.

I suggest improving the manuscript starting to abstract to show the results, conclusion and significance of the study.

Main criticisms:

-In paragraph 2.2., the authors assert that their study has demonstrated that ultrasonic-assisted ethanol extraction scavenging FRAP, DPPH, ABTS and chelating metal ions. This study has demonstrated that ultrasonic-assisted ethanol extraction may be a great efficiency proposal for betulin active compounds from white birch bark but they do not describe the ultrasound parameters used for extraction.

-The chemical characterization of the betulin extract by HPLC is not complete. Figure 3, B shows the majority peak that identifies betulin. However, the other components of the extract, albeit in a smaller amount, should be identified. Indeed, the authors attribute the biological effect of the extract to the presence of betulin, but they did not evaluate whether a possible additive or synergistic effect of the different components of the extract occurs. Also, to attribute the biological activity to betulin, the pure molecule at the same concentration present in the extract should be tested on the cells.

-The effect of betulin extract on HaCaT cells viability is reported in Figure 5. However, the effect of the solvent without the extract should be reported in the figure as control. Moreover, the statistical significance is not shown in the figure.

-The evaluation of the in vitro antioxidant activity of betulin extract does not indicate a potential antioxidant effect of the extract in cellular and/or animal models, thus proving useless for biological purposes and potential use “in the treatment of diseases” as reported by the authors in line 349. The measurement of reactive intracellular oxygen species after treatment with the botulin extract should be performed to demonstrate the antioxidant effect on the HaCaT cells. Furthermore, it could be interesting to evaluate if the reduction of cell viability could be due to a potential modulation of the cell oxidative state by the betulin extract.

Line 84: correct amyl ethanol with amyl alcohol.

Line 204: eliminate “was” after betulin.

Line 207: insert space after release.

Line 419, 459, 461, 470: insert final point.